# Are Artificial Intelligence-Assisted Three-Dimensional Histological Reconstructions Reliable for the Assessment of Trabecular Microarchitecture?

**DOI:** 10.3390/jcm13041106

**Published:** 2024-02-15

**Authors:** János Báskay, Dorottya Pénzes, Endre Kontsek, Adrián Pesti, András Kiss, Bruna Katherine Guimarães Carvalho, Miklós Szócska, Bence Tamás Szabó, Csaba Dobó-Nagy, Dániel Csete, Attila Mócsai, Orsolya Németh, Péter Pollner, Eitan Mijiritsky, Márton Kivovics

**Affiliations:** 1Data-Driven Health Division of National Laboratory for Health Security, Health Services Management Training Centre, Semmelweis University, Kútvölgyi út 2, 1125 Budapest, Hungary; baskay.janos@emk.semmelweis.hu (J.B.); szocska.miklos@semmelweis.hu (M.S.); pollner.peter@emk.semmelweis.hu (P.P.); 2Department of Biological Physics, Eötvös Loránd University, Pázmány Péter Sétány 1/a, 1117 Budapest, Hungary; 3Department of Community Dentistry, Semmelweis University, Szentkirályi Utca 40, 1088 Budapest, Hungary; penzes.dorottya@semmelweis.hu (D.P.); brunaguimaraess@icloud.com (B.K.G.C.); nemeth.orsolya@semmelweis.hu (O.N.); 4Department of Pathology, Forensic and Insurance Medicine, Semmelweis University, Üllői út 93, 1091 Budapest, Hungary; kontsek.endre@semmelweis.hu (E.K.); pesti.adrian@semmelweis.hu (A.P.); kiss.andras@semmelweis.hu (A.K.); 5Department of Oral Diagnostics, Semmelweis University, Szentkirályi Utca 47, 1088 Budapest, Hungary; szabo.bence.tamas@semmelweis.hu (B.T.S.); dobo-nagy.csaba@semmelweis.hu (C.D.-N.); 6Department of Physiology, Semmelweis University, Tűzoltó u. 34-37, 1094 Budapest, Hungary; csete.daniel@semmelweis.hu (D.C.); mocsai.attila@semmelweis.hu (A.M.); 7Department of Head and Neck Surgery and Maxillofacial Surgery, Tel-Aviv Sourasky Medical Center, School of Medicine, Tel Aviv University, Tel Aviv 64239, Israel; mijiritsky@bezeqint.net; 8Goldschleger School of Dental Medicine, Faculty of Medicine, Tel Aviv University, Tel Aviv 39040, Israel

**Keywords:** artificial intelligence (AI), microCT, histomorphometry, three-dimensional histological reconstruction, bone augmentation, dental implant

## Abstract

**Objectives:** This study aimed to create a three-dimensional histological reconstruction through the AI-assisted classification of tissues and the alignment of serial sections. The secondary aim was to evaluate if the novel technique for histological reconstruction accurately replicated the trabecular microarchitecture of bone. This was performed by conducting micromorphometric measurements on the reconstruction and comparing the results obtained with those of microCT reconstructions. **Methods:** A bone biopsy sample was harvested upon re-entry following sinus floor augmentation. Following microCT scanning and histological processing, a modified version of the U-Net architecture was trained to categorize tissues on the sections. Detector-free local feature matching with transformers was used to create the histological reconstruction. The micromorphometric parameters were calculated using Bruker’s CTAn software (version 1.18.8.0, Bruker, Kontich, Belgium) for both histological and microCT datasets. **Results:** Correlation coefficients calculated between the micromorphometric parameters measured on the microCT and histological reconstruction suggest a strong linear relationship between the two with *p*-values of 0.777, 0.717, 0.705, 0.666, and 0.687 for BV/TV, BS/TV, Tb.Pf Tb.Th, and Tb.Sp, respectively. Bland–Altman and mountain plots suggest good agreement between BV/TV measurements on the two reconstruction methods. **Conclusions:** This novel method for three-dimensional histological reconstruction provides researchers with a tool that enables the assessment of accurate trabecular microarchitecture and histological information simultaneously.

## 1. Introduction

Implant-borne prostheses are a safe and predictable option in oral rehabilitation [1]. Several bone microarchitectural factors have been associated with the failure of osseointegration, such as low bone quality, thin cortical bone, and sparse trabecular bone [2,3].

It has been established that poor recipient bone quality is an important risk factor in early implant failure [4,5,6]. Bone quality assessment prior to or at the time of implant placement influences the clinician’s decision-making regarding the choice between submerged or non-submerged implant placement and determines the protocol for prosthetic loading [7,8,9]. Therefore, predicting bone quality at dental implant recipient sites prior to surgery is essential for the clinician. Postoperatively, microarchitectural evaluation of dental implant recipient bone through histological and microcomputed tomography (microCT) analysis of bone core biopsy samples allows the researcher to determine the risk factors involved in implant failure and evaluate the clinical performance of bone graft materials. Histological examination of bone core biopsy samples harvested during dental implant placement may help identify underlying pathologies in cases of implant failure [10,11].

There is no consensus in the literature regarding the exact definition of bone quality [12]. The term bone quality incorporates the degree of mineralization, cortical bone thickness [13], and trabecular bone morphology [14,15]. Hounsfield unit (HU) measurements on computed tomography (CT) reconstructions, insertion torque resistance, and resonance frequency analysis (RFA) are the most prevalent methods to evaluate bone quality before and during implant placement [7,8,16,17,18,19]. Cone beam computed tomography (CBCT) has become the standard imaging method in the dento-maxillofacial region for the preoperative assessment of dental implant recipient bone anatomy and quantity. However, gray-level measurements performed on CBCT reconstructions are not reliable for the assessment of bone quality [20,21]. MicroCT analysis has been successfully applied to assess the microarchitecture of bone core biopsy samples obtained from the implant bed to evaluate the bone quality of dental implant recipient bone following surgery [6,10,13,16,22,23,24,25,26]. Studies show that the results of micromorphometric analysis conducted with microCT correlate highly with histomorphometric measurements, with it being considered as the gold standard for the structural analysis of trabecular bone and implant osseointegration [25,26,27,28]. However, microCT analysis may be more reliable in cases where micromorphometric parameters are to be calculated to assess the true three-dimensional microarchitecture [22,23,24,29]. Conventional histomorphometry provides information on tissues and cellularity [30]. Nevertheless, histomorphometry is usually carried out on representative slices and provides little information on three-dimensional structures [22]. MicroCT is a non-destructive modality to evaluate the three-dimensional structure of trabecular bone in high resolution [31,32]. However, microCT reconstruction is based on the radiodensity of structures and provides no information on cellularity. Manual or semiautomatic thresholding is the most widespread method to differentiate between bone and soft tissue.

Recently, artificial intelligence (AI) has been applied for pathological and medical image analysis with adequate sensitivity and specificity [33,34]. In dentistry, AI has been successfully applied for caries diagnostics, the detection of apical periodontitis and odontogenic cysts, orthodontic diagnosis and treatment planning, the identification of the inferior alveolar nerve, the segmentation of teeth and jaw bones, airway modeling, sinus mucosa thickening, dental implant type recognition, and degenerative changes in the temporomandibular joint [35,36,37]. The use of AI in image annotation enables fast and reliable differentiation between tissues in large quantities on pathologic images, decreasing the risk of fatigue-related errors [38,39]. However, oversight of a professional may be desirable [40,41].

Various methods have been introduced for the three-dimensional reconstruction of serial sections [42,43,44]. Following the three-dimensional reconstruction of the serial sections, micromorphometric parameters—such as in the case of microCT reconstructions—may be calculated. In this case, distinguishing between different types of tissues is more reliable than radiological thresholding [45]. However, the three-dimensional reconstruction of serial sections c [43]. A three-dimensional histological reconstruction that represents the true trabecular bone microarchitecture would allow for a more comprehensive analysis of implant recipient bone. Such an assay may be utilized in clinical research on bone graft materials and the clinical pathology of implant failure as well.

To the best of our knowledge, there have been no studies conducted to validate micromorphometric measurements carried out on three-dimensional histological reconstructions using microCT analysis.

The aim of this study was to create a three-dimensional histological reconstruction through the AI-assisted classification of tissues and alignment of the serial sections. The secondary purpose of this study was to assess whether the novel, AI-assisted method for three-dimensional histological reconstruction reproduces the true trabecular microarchitecture of bone by performing micromorphometric measurements on the three-dimensional histological reconstructions and comparing their results to those carried out on microCT reconstructions.

## 2. Materials and Methods

The study was approved by the Medical Research Council Committee of Science and Research Ethics (ETT TUKEB BM/18442-1/2023), and it was conducted in accordance with the Declaration of Helsinki. The interventions carried out during this study were thoroughly explained to the patient enrolled. The patient signed the necessary informed consent documents.

### 2.1. Surgical Procedures

A 50-year-old female patient presented to our Department of Community Dentistry with a chief complaint of the missing upper left first molar. An implant-borne crown was planned to substitute the missing tooth. The patient did not report uncontrolled medical disorders, systemic diseases, or medication that would alter bone metabolism. The preoperative CBCT scan revealed a fully healed site with a residual bone height of 3.2 mm below the maxillary sinus and an alveolar ridge of 7.5 mm in width. Sinus floor elevation (SFE) was proposed to restore the sufficient vertical dimensions of the recipient bone.

The patient rinsed with a 0.2% chlorhexidine solution before surgery. Under local anesthesia, a full-thickness flap was elevated from a crestal incision with a mesial and distal releasing incision. Lateral window osteotomies were carried out using a piezoelectric surgery device (NSK Variosurg3 Ultrasonic Bone Surgery System, NSK Europe GmbH, Eschborn, Germany) and the Schneiderian membrane was carefully elevated. A bovine xenograft (Cerabone Granulate S, Botiss biomaterials GmbH, Zossen, Germany) was packed in the base of the sinus with light force. The lateral window was sealed using a collagen membrane (Jason membrane, Botiss biomaterials GmbH, Zossen, Germany). The flap was mobilized using a periosteal incision to achieve tension-free primary closure. The flap was closed with single interrupted sutures. Suture removal took place after 10 days. The patient received 1 g of amoxicillin–clavulanate (Aktil Duo 875 mg/125 mg, Sandoz Hungária Kft., Budapest, Hungary) twice per day, starting on the day of the surgery. A non-steroid anti-inflammatory drug, diclofenac (Cataflam 50 mg, Novartis Hungária Kft., Budapest, Hungary), 3 times a day for 3 days was prescribed to manage postoperative pain. The patient used a 0.2% chlorhexidine mouth rinse (Corsodyl, GlaxoSmithKline Consumer Healthcare GmbH & Co. KG, München, Germany), twice a day for 2 weeks. The patient did not wear a temporary prosthesis during the healing period, which lasted 6 months. During surgical re-entry, implant bed preparation was carried out with a trephine drill with an external diameter of 3.5 mm and an internal diameter of 2.5 mm (330 205 486 001 025 Hager & Meisinger GmbH, Neuss, Germany) at 800 rpm to collect a bone core biopsy sample from the augmented bone for histologic and microCT analysis. The implant bed was finalized according to the instructions of the implant manufacturer. A dental implant (Denti Root Form, Denti System Ltd., Szentes, Hungary) was inserted non-submerged in the augmented bone. The healing period was 3 months before the implant prosthetic procedure.

The bone core biopsy sample was placed in 4% formaldehyde solution in 0.1 M phosphate-buffered saline (PBS), pH 7.3, and was stored at 4 °C.

### 2.2. MicroCT Imaging

The bone core biopsy sample was scanned using a SkyScan 1272 microCT scanner (Bruker, Kontich, Belgium) with a voxel size of 11 µm at 60 kV, 66 mA. A 0.25 mm aluminum filter was used to decrease image noise, and a rotation step of 0.5° was set prior to image acquisition. After scanning, reconstruction of raw images was performed using SkyScan NRecon software (version 2.0, Bruker, Kontich, Belgium).

### 2.3. Histological Processing and Scanning

The biopsy sample was decalcified for 4 days and then embedded in paraffin (FFPE). The horizontal embedding orientation was obtained by laying the biopsy between tissue foam pads. Approximately 5 µm thick serial sections were cut. Sections picked up on standard glass slides were routinely stained with hematoxylin and eosin (H&E) then covered and scanned using a 3Dhistech Panoramic 1000 Digital Slide Scanner (3Dhistech, Budapest, Hungary).

### 2.4. Three-Dimensional Histological Reconstruction

To establish a structural ground truth for the histology reconstruction, the corresponding microCT volume was sliced in silico in the same plane as the real sample. To ensure the convergence of the histological reconstruction process, torn, damaged, or otherwise visually distorted slides were discarded. The reconstruction process has two main steps: pre-alignment using data from the histology reconstruction and fine alignment with the help of the microCT volume.

The MIRAX slide scanning process produces high-resolution images with variable dimensions, typically measuring 100,000 × 200,000 pixels with a pixel ratio of 0.121267 µm. To prepare these slides for further analysis, a pre-processing step was performed using the OpenSlide [46] and OpenCV [47] software libraries.

The pre-processing pipeline involves the detection of the region of interest (ROI), which is critical for possible downstream morphometry analysis. The pipeline starts with gray-scale conversion and Canny edge detection, with the parameters ‘threshold1 = 100’ and ‘threshold2 = 255’. An adaptive thresholding algorithm with the parameters ‘maxValue = 255, adaptiveMethod = 1, thresholdType = 1, blockSize = 11, and C = 2’ is then applied. The resulting binary image is subjected to 15 iterations of dilation and erosion to reduce noise and ensure smoother contours. Contour detection is then used to identify the boundaries of the ROI. The bounding rectangle for each contour is calculated, and the largest of these is considered to contain the contour of the biopsy. The biopsy’s contour is then fitted with the minimum area (“rotated”) rectangle, as well as a concave polygon to determine the ROI for morphometric analysis.

ROI detection is performed on the highest zoom level (lowest resolution) to achieve optimal performance. The rotated rectangle and concave polygons obtained at this stage are then scaled to the desired zoom level using bicubic scaling followed by simple thresholding (‘threshold = 128’) to remove artifacts and produce binary images. The angle of the rotated rectangle is used to calculate the rotation required for coarse pre-alignment.

The resulting images are exported at a resolution of 3000 × 7000, which corresponds to an 8.0× downsample from the lowest zoom level of the original slide, with a pixel ratio of 0.970136 µm. The downsampling factor can be decreased or eliminated, but this would result in a significant increase in computation time for downstream tasks. Each slice is fitted with a minimal area covering the rectangle, which when rotated upright is responsible for the coarse alignment of the slice rotations. This is followed by a stepwise affine transformation described by Nagara et al. [48]. The process begins at one end of the stack of slices and iteratively transforms the entire stack to match the other end of the slices. Within this iterative transformation, the reference slices were chosen based on the number of keypoints detected by the deep feature matching algorithm, detector-free local feature matching with transformers (LoFTR) [44]. LoFTR uses a convolutional neural network (CNN) backbone to compute feature descriptors for the images, which are then probed for matches using a transformer model. Once the reference slice is identified, the optimal affine transformation, which minimizes the cross-entropy between the images is calculated by first finding the optimal translation, followed by the optimal rotation and then finally the optimal scaling. By separating the different elements of the affine transformation, a more stable result can be achieved, and false optima can be avoided.

The final alignment matches each histology slide with one of the in-silico microCT slices. The reference slices within the microCT volume are found by calculating the number of matching LoFTR keypoints for each in-silico microCT slice for a given histology slide. We found that the number of matching keypoints for the corresponding slides shows a peak that is an order of magnitude higher than the background. To properly identify the location of this peak, a bell curve is fitted, and its mean is used to determine the maximum; this process helps reduce the effects of noise within the keypoint distribution. Once the match is identified, the calculation of the optimal affine transformation follows the same procedure as in the pre-alignment step. The matching process is presented in Figure 1 and Figure 2.

The beneficial side-effect of using microCT as a structural ground truth is that the resulting 3D histology is already registered to the microCT volume, which enables downstream comparisons.

### 2.5. Tissue Segmentation

Ground truth annotations were created for seven slides using QuPath software (version 0.1.2, Queen’s University Belfast, Northern Ireland, UK) to differentiate between three histological categories: bone tissue, residual bone graft material, and non-mineralized tissue [49]. The three categories were separated based on their distinct morphological features. To address the imbalanced nature of the segmentation problem, class weights were calculated based on the ground truth of the training set.

These maps were split into tiles of 512 × 512 pixels in size with a 64-pixel overlap in each direction and then divided into a five-fold cross-validation setup, meaning that, within each fold, the model was trained on 80% of the data and validated on 20%. U-Net architecture [45] was trained for the semantic segmentation task, modified from the original to include batch normalization in the 3 × 3 2D convolution blocks and dropout after the concatenations, to combat overfitting and improve the generalizability with limited training data, which was confirmed by the resulting validation accuracies being 0.9548, 0.9565, 0.9304, 0.9608, 0.9607 for the five folds, respectively. The exact architecture is depicted in Figure 3. The model was compiled using the Adam optimizer, categorical cross-entropy loss function, a batch size of 16, and a 5% dropout rate and trained for 200 epochs with a learning rate schedule that started from 1 × 10^−3^ and decreased by a factor of 10 down to 1 × 10^−5^ if the validation loss did not improve for 30 epochs on a single NVIDA V100 32 GB GPU. The segmentation maps were then obtained by ensembling the five models trained on the five folds, producing a three-component vector for each pixel on the input image, where each component corresponded to a probability for one of the classes. The results were reconstructed by averaging the overlaps and were then subjected to bilateral filtering with a filter size of 32 pixels and standard deviation in both color and a pixel space of 64 to remove neural network artifacts. It is advantageous to use bilateral filtering instead of Gaussian filtering for noise elimination, since the latter blurs the sharp edges, while the former does not due to it also applying a filter in the “color space” as well as in the “real space”. The segmentation maps were thresholded so that each pixel belonged to a single, most probable class. An example from the segmentation process is presented in Figure 4.

The three-dimensional structure of the bone tissue and bone graft material was recovered by applying the same transformations calculated in the 3D reconstruction step described in Section 2.4.

### 2.6. Measurement Pre-Processing

To obtain a reasonably large statistic for comparison, both the microCT and the three-dimensional histological reconstruction were tiled into 512 × 512 × 230 micron sized matching rectangular prisms, resulting in 47 non-overlapping samples.

### 2.7. Micromorphometric Analysis of the MicroCT and Three-Dimensional Histological Reconstructions

The micromorphometric parameters were calculated using CTAn software (version 1.18.8.0, Bruker, Kontich, Belgium) for both histological and microCT datasets by applying the ROI mask and semi-automatic binary selection and choosing 3D analysis in the morphometry view. In addition to basic parameters (tissue volume (TV), bone volume (BV), percent bone volume, tissue surface (TS), bone surface (BS), BS/BV ratio, bone surface density (BS/TV), and centroid and moments of inertia), all additional values were also calculated: structure model index (SMI), trabecular pattern factor (Tb.Pf), trabecular thickness (Tb.Th), trabecular number (Tb.N), trabecular separation (Tb.Sp), degree of anisotropy (DA), fractal dimension (FD), number of objects (Obj.N), number of closed pores (Po.N(cl)), porosity (Po), and Euler number (EuN).

### 2.8. Statistical Analysis

#### 2.8.1. Spearman’s Rank Correlation Coefficient

To compare the results of the micromorphometric measurements, Spearman’s rank correlation coefficient (⍴) was used, as implemented by SciPy [50]. The correlation coefficient measures the monotonic relationship between two variables and takes values between −1 and 1, with the extremes meaning that one variable is a perfectly monotonic function of the other, with the *p*-value corresponding to the hypothesis that the two variables are linearly uncorrelated. Values of *p* < 0.05 were considered significant.

#### 2.8.2. Bland–Altman and Mountain Plots

To visualize the agreement between the micromorphometric measurements on the two reconstruction methods for a given parameter, the Bland–Altman and mountain plots were calculated.

The Bland–Altman plot was constructed by assigning the mean of measurement for the x value and the difference to the y value [51]. Then, the bias was obtained from the mean of differences, and the 95% limits of agreement were obtained by multiplying the standard deviation of the differences with ± 1.96, given that the differences are normally distributed. It was also suggested by Altman and Bland that a one-sample *t*-test for zero bias and linear regression between the means and differences should be performed to test for zero slope.

Mountain plots were obtained by calculating the cumulative density function of the differences and folding them at the median, by taking 1-*p*, if *p* > 0.5, and they were used to complement the difference plots [52].

## 3. Results

Correlation coefficients calculated between the micromorphometric parameters measured on the microCT image tiles and the tiled 3D reconstruction suggested a strong linear relationship between the two with ⍴-values of 0.777, 0.717, 0.705, 0.666, and 0.687 for BV/TV, BS/TV, Tb.Pf Tb.Th, and Tb.Sp, respectively, and an average ⍴-value across all measured values of 0.605. This was further supported by consistently small (~1 × 10^−8^) p-values, suggesting that the high Spearman correlation was not due to random chance.

The Bland–Altman and mountain plots in Figure 5 were only constructed for the five previously mentioned parameters.

The statistical tests for normally distributed differences yielded results that cannot reject this hypothesis. The t-tests for zero bias were significant for percent bone volume. A linear regression was performed on the means and differences, also suggesting zero slope for percent bone volume. Statistical tests and their results are presented in Table 1.

## 4. Discussion

According to the results of the present study, a strong and statistically significant linear correlation was observed between the micromorphometric variables calculated from the three-dimensional histological reconstruction and microCT datasets. The Bland–Altman and mountain plots show sufficient agreement between BV/TV calculated from the microCT and three-dimensional histological reconstruction datasets. BV/TV is considered the most important outcome measure among the micromorphometric parameters as it corresponds with the percent bone area calculated in the histomorphometric analysis [10]. However, the results of Bland–Altman and mountain plots are less ideal when evaluating the agreement between the micromorphometric parameters related to surface measures (BS/BV, Tb.Pf, Tb.Th, and Tb.Sp) between the two three-dimensional datasets. In this study, the microCT reconstructions consisted of isotropic voxels, whereas the voxels of the three-dimensional histological reconstruction were anisotropic. The thickness of the slides was set at 5 µm, while the resolution of the mask following the classification of different tissues was higher. This may have led to disagreement in the surface-related micromorphometric parameters between the microCT and three-dimensional histological reconstruction datasets.

In clinical practice, it is essential to predict bone quality before implant placement to assess the risk of implant failure and to determine the time of prosthetic loading [7,8,9]. CBCT has become a standard for three-dimensional imaging modalities in the dento-maxillofacial region prior to implant placement and bone augmentation because of its low radiation dose compared to CT and its high spatial resolution [20,53,54]. However, unlike Hounsfield units (HUs) measured on CT reconstructions, gray-level measurements on CBCT are unreliable and should be avoided [20,21]. Micromorphometric parameters calculated from CBCT reconstructions may be reliable indicators of bone quality, as the literature suggests that these measurements correlate well with those, calculated from microCT reconstructions of bone core biopsy samples harvested from the ROI [11,55,56,57,58,59,60,61]. However, a correlation between the two variables does not infer a valid method for measurement. Because of these difficulties in predicting bone quality, it is increasingly important to research recipient bone microarchitecture following bone augmentation procedures to determine whether factors such as healing time, bone graft materials used, defect volume, the characteristics of the local anatomy, and systemic diseases or medication influence the quality of augmented bone.

Assessment of trabecular bone microarchitecture is pivotal in bone augmentation and dental implant-related research. Histomorphometry is considered the gold standard of bone quality assessment [12,30,62]. The examiner calculates the percentage of newly formed bone, bone marrow, and residual graft material from representative sections, which are widely accepted as short-term outcome measures for the integration of biomaterials in the augmented area. However, histomorphometry only allows for assessment in two dimensions, which is considered a significant limitation [22]. The results of histological measurements depend on the slice thickness and cutting direction of the histological sections [23,24,29]. Three-dimensional analysis of trabecular bone microarchitecture is important for the comprehensive assessment of bone quality. Serial sectioning, digitization of the sections, segmentation of the tissue types identified within the biopsy sample, and three-dimensional reconstruction of the sections enable analysis of trabecular microarchitecture. Convolutional neural networks enable quick and reliable classification of different tissue types on histological sections and assist three-dimensional reconstruction of serial sections [33,41]. However, artifacts may cause bias in such a three-dimensional reconstruction [43]. Therefore, validation of the method presented is necessary with a modality that enables direct analysis of trabecular microarchitecture.

MicroCT enables direct assessment of the three-dimensional trabecular structure in high resolution based on the radiodensity of the different tissues found in the biopsy samples. It is non-destructive and allows for further processing of the bone core biopsy sample for histology or biomechanical assays. A lack of information on tissues and cellularity is a limitation of microCT analysis [10,12,28]. Nevertheless, microarchitectural analysis of three-dimensional reconstructions of serial sections may combine the advantages of histomorphometric and microCT analysis, providing micromorphometric data as well as information on tissues and cellularity. Such a three-dimensional histological reconstruction can be used in clinical and preclinical biomaterial research for the postoperative evaluation of augmented bone. For the clinician, a comprehensive microarchitectural and histological assessment of implant recipient bone following dental implant placement may identify potential causes of implant failure.

Because of the time-consuming demineralization, serial sectioning, and staining involved, the three-dimensional histological reconstruction method described in our study is not applicable as an intraoperative method to assess bone quality. Therefore, it may be used primarily in research. Another limitation of the method is its destructive nature, which precludes further testing of the biopsy specimen. A further limitation of the present study is that it was carried out in a single biopsy specimen and only three tissues (newly formed bone, residual graft material, and non-mineralized tissues) were differentiated within the histological sections. Further studies conducted with the three-dimensional histological reconstruction method described in our study to assess vascular structures and bone remodeling markers may be of interest. Another avenue of research may be the feasibility of the novel method of three-dimensional histological reconstruction in the assessment of soft tissues.

## 5. Conclusions

In this study, a three-dimensional histological reconstruction through the AI-assisted classification of tissues and alignment of the serial sections was described. According to the results of the present study, a strong and statistically significant correlation was observed between the micromorphometric variables calculated from the three-dimensional histological reconstruction and microCT datasets.

Thus, the novel method for the three-dimensional reconstruction of histological sections described in our study provides researchers with a tool that enables the accurate assessment of three-dimensional trabecular microarchitecture and histological information simultaneously.

## Figures and Tables

**Figure 1 jcm-13-01106-f001:**
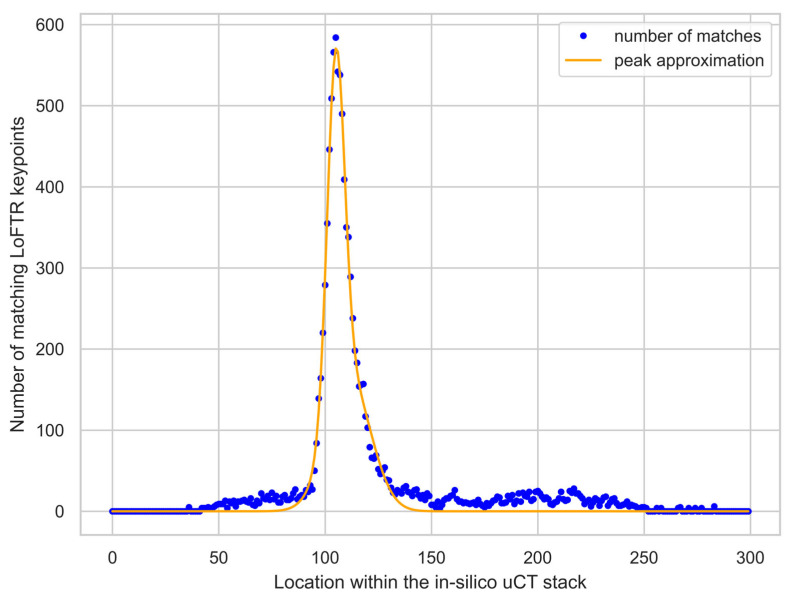
Example of the matching process within the final alignment. The number of matching keypoints for each in-silico microCT slice is represented by a blue dot. Notice that the slice that corresponds to the histological slide used in this example has an order of magnitude more matching keypoints then the rest of the microCT slices. This peak in the keypoint curve was approximated using a bell curve, represented by the orange line, the mean of which corresponds to the in-silico microCT slice matching the histology slice.

**Figure 2 jcm-13-01106-f002:**
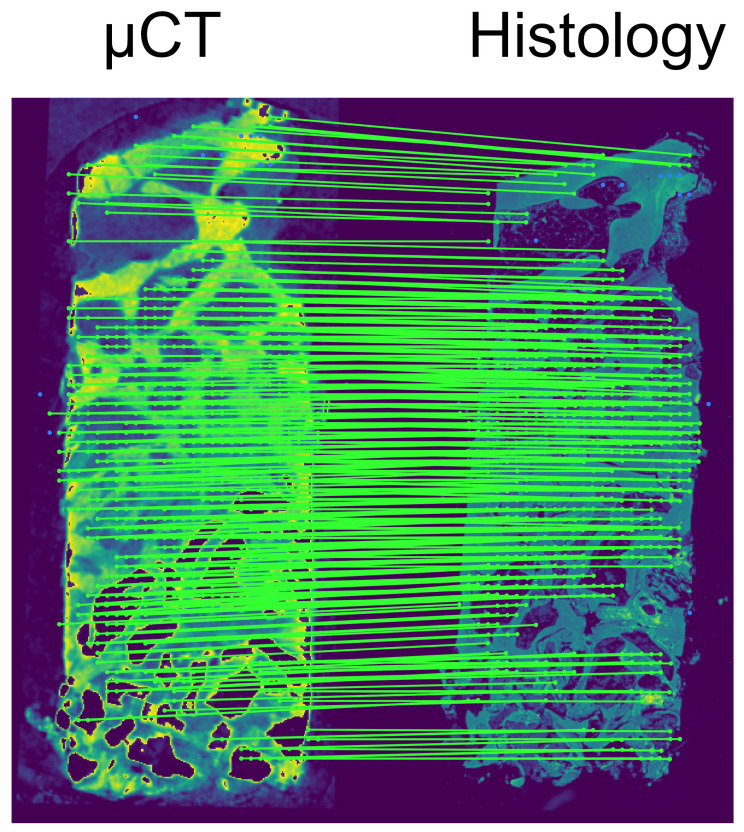
Visualization of the matched LoFTR keypoints between the two imaging modalities: an in silico microCT slice and the histology; this specific match corresponds to the peak of the curve presented in Figure 1. Notice that there are only parallel lines connecting the keypoints; crossing lines would indicate false matches.

**Figure 3 jcm-13-01106-f003:**
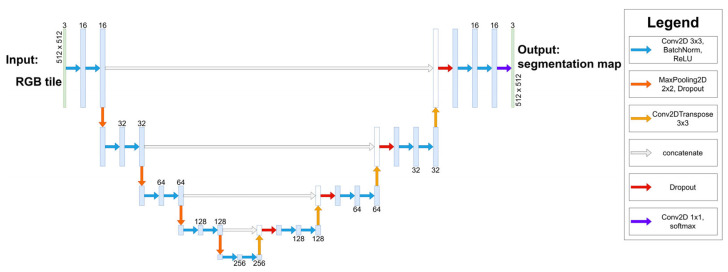
U-Net architecture used in tissue segmentation. Since each 2 × 2 MaxPooling layer halves the resolution of the feature map, the resolution of the bottleneck layer is 32 × 32 pixels. The blue boxes correspond to multi-channel feature maps with the number of channels noted either above or below the boxes. The white boxes represent the concatenation between the upsampled (Conv2DTransposed) feature map from the expanding part and the copied feature map from the contracting path.

**Figure 4 jcm-13-01106-f004:**
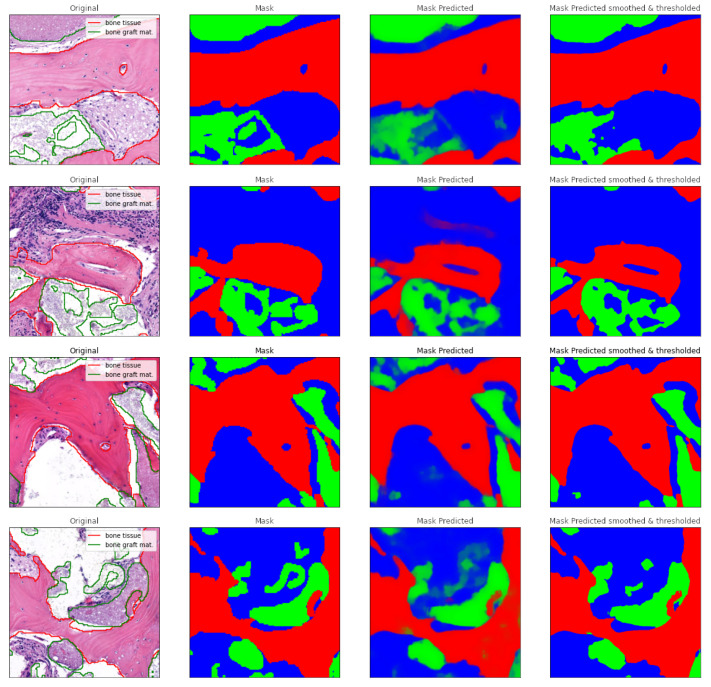
Side-by-side examples showing the inputs and outputs for the tissue segmentation model. The first column, named “Original”, depicts the tile from the histology, overlayed with the handmade annotations (bone outlined with red and bone graft material outlined with green). The second column, named “Mask”, shows the annotations as segmentation masks. These masks delineate three histological categories—bone tissue (highlighted in red), bone graft material (in green), and non-mineralized tissue (in blue). The third column, named “Mask Predicted”, depicts examples of the U-Net model’s outputs, with paler colors indicating smaller probabilities for a given class. The fourth column, named “Mask Predicted smoothed & thresholded”, depicts the final annotation following the post-processing consisting of bilateral filtering (”smoothing”) and thresholding, which makes each pixel belong to the single most probable class only.

**Figure 5 jcm-13-01106-f005:**
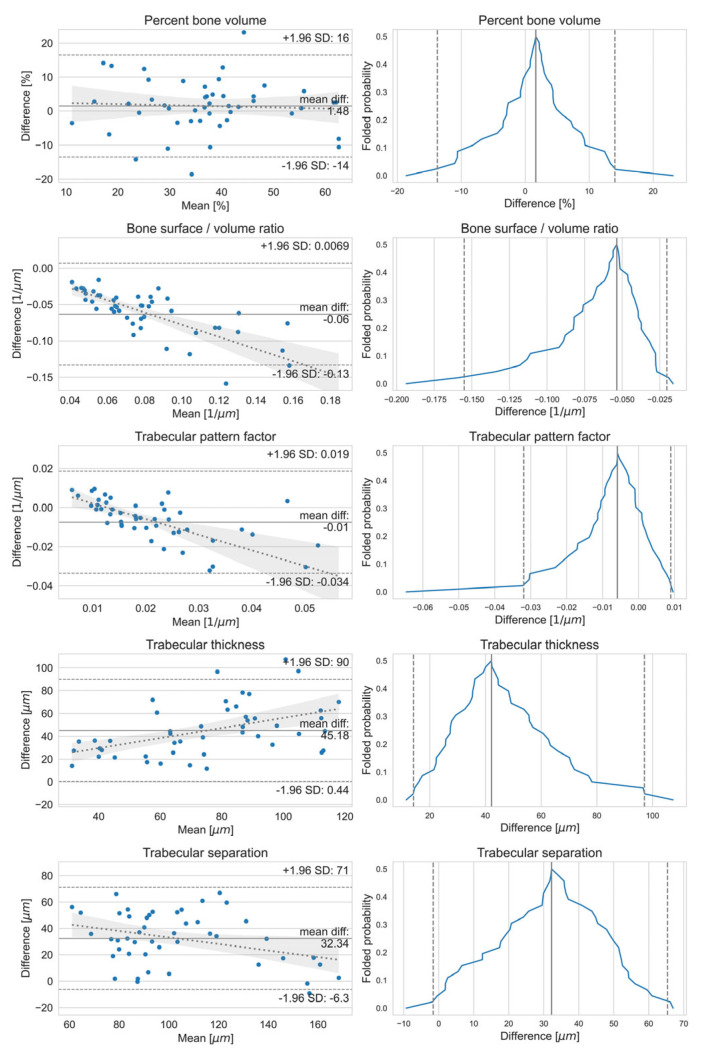
Bland-Altman (BA) plots on the left and mountain plots on the right comparing the microCT tiles and the 3D histological reconstruction tiles for the measured micromorphometrical parameters percent bone volume, bone surface/bone volume ratio, trabecular pattern factor, trabecular thickness, and trabecular separation. On the BA plots, the mean difference is indicated by a solid line, the 95% limits of agreements are indicated by dashed lines, and the fitted linear regression is indicated by a dotted line along with its 95% confidence band. Each parameter has close to zero bias (the slope of the mean–difference curve), but further statistical tests showed that this hypothesis cannot be rejected only for the percent bone volume; for the other parameters, the difference from zero bias is significant. On the mountain plots, the median difference is marked with a solid line and the 95% confidence interval is marked with dashed lines. The median difference indicates the bias between the two imaging methods; the closer it is to zero, the smaller the bias. Only the trabecular thickness measurements show a larger bias; however, all distributions show fairly long tails.

**Table 1 jcm-13-01106-t001:** Statistical tests were performed to compare five micromorphometric parameters: BV/TV, BS/TV, Tb.Pf, Tb.Th, and Tb.Sp., measured using microCT and 3D histological reconstruction. The normality test assesses the null hypothesis that the differences follow a normal distribution; the obtained results are not significant. The one-sample *t*-test evaluates whether the bias is zero. For BV/TV, the test is inconclusive, but for the rest of the parameters, this hypothesis must be discarded. Linear regression was used to test for a zero slope between the differences and means, which only holds for BV/TV. Spearman’s rank correlation coefficient quantifies whether there is a strong linear relationship between the values. This is true for all micromorphometric parameters, with small accompanying *p*-values indicating statistical significance.

	Normality Test		One Sample *t*-Test		Linear Regression			Spearman Correlation	
	Statistic	*p*	Statistic	*p*	Slope	Intercept	*p*	Correlation Coefficient	*p*
BV/TV	2.160	0.340	1.313	0.196	−0.033 ± 0.091	2.691 ± 3.556	0.721	0.777	1.389 × 10^−10^
BS/TV	22.399	1.000	−11.977	9.971 × 10^−16^	−0.839 ± 0.095	−0.006 ± 0.009	2.291 × 10^−11^	0.717	1.441 × 10^−8^
Tb.Pf	31.800	1.000	−3.824	3.933 × 10^−4^	−0.800 ± 0.116	0.010 ± 0.003	1.436 × 10^−8^	0.705	3.187 × 10^−8^
Tb.Th	5.364	0.068	13.423	1.608 × 10^−17^	0.441 ± 0.122	11.889 ± 9.635	7.118 × 10^−4^	0.666	3.272 × 10^−7^
Tb.Sp	3.483	0.175	11.118	1.257 × 10^−14^	−0.247 ± 0.102	57.799 ± 10.844	0.019	0.687	9.561 × 10^−8^

## Data Availability

Data are accessible at: Baskay J, Kivovics M, Penzes D, Kontsek E, Pesti A, Szocska M, Nemeth O, and Pollner P. 2022. Reconstructing 3D histological structures using machine learning (AI) algorithms. IDR Tissue Archive. https://doi.org/10.17867/10000184 (accessed on 4 February 2024).

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
