# Peer review of "Are Artificial Intelligence-Assisted Three-Dimensional Histological Reconstructions Reliable for the Assessment of Trabecular Microarchitecture?"

_jcm, 2024, doi:10.3390/jcm13041106_

Round 1
Reviewer 1 Report
Comments and Suggestions for Authors
dear authors
first of all thank you for your contributions, I have read the draft with great interest.
Nevertheless I would recommend to modify the paper as follows:
- Introduction: please expand the analysis of the dental literature investigating the usage of Artificial intelligence in the same field. What is known? what is missing?
- Discussion section: It is not clear what kind of clinical implications the research might deal with. please improve the discussion section and the conclusions accordingly.
Author Response
Point to point response to Reviewer 1
Reviewer 1 (R1): dear authors first of all thank you for your contributions, I have read the draft with great interest.
Corresponding Author (CA): We would like to thank the reviewer their time and effort to identify the shortcomings of our manuscript.
R1: Nevertheless I would recommend to modify the paper as follows:
- Introduction: please expand the analysis of the dental literature investigating the usage of Artificial intelligence in the same field. What is known? what is missing?
CA: We have added the following to the Introduction section:
“In dentistry, AI has been successfully applied for caries diagnostics, detection of apical periodontitis and odontogenic cysts, orthodontic diagnosis and treatment planning, identification of the inferior alveolar nerve, segmentation of teeth and jaw bones, air-way modelling, sinus mucosa thickening, dental implant type recognition, and degen-erative changes of the temporomandibular joint.”
citing the following references:
- Mureșanu, S.; Almășan, O.; Hedeșiu, M.; Dioșan, L.; Dinu, C.; Jacobs, R. Artificial intelligence models for clinical usage in dentistry with a focus on dentomaxillofacial cbct: A systematic review. Oral Radiol 2023, 39, 18-40.
- Mohammad-Rahimi, H.; Motamedian, S.R.; Rohban, M.H.; Krois, J.; Uribe, S.E.; Mahmoudinia, E.; Rokhshad, R.; Nadimi, M.; Schwendicke, F. Deep learning for caries detection: A systematic review. Journal of dentistry 2022, 122, 104115.
- Revilla-León, M.; Gómez-Polo, M.; Vyas, S.; Barmak, B.A.; Galluci, G.O.; Att, W.; Krishnamurthy, V.R. Artificial intelligence applications in implant dentistry: A systematic review. The Journal of prosthetic dentistry 2023, 129, 293-300.
R1: - Discussion section: It is not clear what kind of clinical implications the research might deal with. please improve the discussion section and the conclusions accordingly.
CA: We have added the following to the Discussion section to identify potential applications of the three-dimensional histological reconstruction:
“However, correlation between two variables does not infer a valid method for measurement. Because of these difficulties in predicting bone quality, it is increasingly important to research recipient bone microarchitecture following bone augmentation procedures to determine whether factors such as healing time, bone graft materials used defect volume, characteristics of local anatomy, and systemic diseases or medication influence quality of augmented bone.”
Such a three-dimensional histological reconstruction can be used in clinical and pre-clinical biomaterial research for postoperative evaluation of augmented bone. For the clinician a comprehensive microarchitectural and histological assessment of implant recipient bone following dental implant placement may identify potential causes of implant failure.”
We would like to thank the reviewer once again for their work and suggestions. Adding the established applications of AI to the Introduction section and explaining potential clinical and research applications of the method for three-dimensional histological reconstruction greatly improves our manuscript. We hope that the reviewer finds the changes made sufficient, and recommends our manuscript for publication.

Reviewer 2 Report
Comments and Suggestions for Authors
This paper focuses on the issues related to bone quality assessment before dental implantation. A novel three-dimensional histological reconstruction method by AI-assisted classification of tissues and alignment of the serial sections was proposed. Then, by comparing to the reconstruction performed on microCT, the novel method was evaluated if it can accurately replicate the true trabecular microarchitecture of bone. The results prove that the novel method enables researchers to assess the accurate trabecular microarchitecture and histological information simultaneously. Overall the research is innovative and well organized. More detailed comments are provided below:
(1) What the orange and blue lines indicate in Figure 1?
(2) Chapter 2.4 and 2.5 lacks detailed descriptions of the LoFTR reconstruction and the modified version of U-Net segmentation methods adopted, which is not easy for readers to understand the 3D reconstruction method and procedure introduced in the paper. It is recommended to provide a short but clear description of the core ideas of the existing methods when quoting it, in order to enhance the readability and coherence of the manuscript.
(3) What is the total amount of data required to train the U-Net model? As far as I understand, the data only involves the one patient mentioned in Chapter 2.1, so how can the diversity of data required to train the model be guaranteed? The paper only divides the training set and testing set, why does the author not consider the conventional train-val-test division, or the author uses cross-validation to train the model to avoid the ideal results brought by data leakage?
(4) The clarity of the figures and the texts in the figures needs to be improved, such as Figure 3 and Figure 4.
(5) Table 1 is incomplete and may need to be reformatted.
(6) Is it a preoperative or intraoperative bone assessment? As far as I understand, Chapter 1 wants to express the importance of preoperative evaluation, but in the experiment, it is an intraoperative evaluation (after the bone biopsy sample was taken, the implant placement procedure was performed). The author is invited to explain this part.
Author Response
Point to point response to Reviewer 2
Reviewer 2 (R2): This paper focuses on the issues related to bone quality assessment before dental implantation. A novel three-dimensional histological reconstruction method by AI-assisted classification of tissues and alignment of the serial sections was proposed. Then, by comparing to the reconstruction performed on microCT, the novel method was evaluated if it can accurately replicate the true trabecular microarchitecture of bone. The results prove that the novel method enables researchers to assess the accurate trabecular microarchitecture and histological information simultaneously. Overall the research is innovative and well organized. More detailed comments are provided below:
Corresponding author (CA): We would like to thank the reviewer for their time and effort to critically evaluate our manuscript and identify issues that need to be addressed.
R2: (1) What the orange and blue lines indicate in Figure 1?
CA: We have updated the manuscript and the figure to improve clarity. What used to be a blue line, now represented by dots, are the matching keypoints within the microCT volume and a given histological slide. The orange curve is a double bell-curve approximation for finding the peak, which corresponds to the ‘layer’ within the microCT volume that is the most similar, and assumed physically the closest to the histological slide, and as such can be used as a ‘structural ground truth’ in the registration process.
R2: (2) Chapter 2.4 and 2.5 lacks detailed descriptions of the LoFTR reconstruction and the modified version of U-Net segmentation methods adopted, which is not easy for readers to understand the 3D reconstruction method and procedure introduced in the paper. It is recommended to provide a short but clear description of the core ideas of the existing methods when quoting it, in order to enhance the readability and coherence of the manuscript.
CA: We have revised sections 2.4 and 2.5 as requested, as well as added a section with certain model parameters and a description of the semantic segmentation network to improve the reproducibility and understandability of the study.
“The MIRAX slide scanning process produces high-resolution images with variable dimensions, typically measuring 100000 x 200000 pixels with a pixel ratio of 0.121267 µm. To prepare these slides for further analysis, a pre-processing step was performed using the OpenSlide [46] and OpenCV [47] software libraries.
The pre-processing pipeline involves the detection of the region of interest (ROI), which is critical for possible downstream morphometry analysis. The pipeline starts with gray-scale conversion and Canny edge detection, with parameters `threshold1 = 100` and `threshold2 = 255`. An adaptive thresholding algorithm with parameters `maxValue = 255, adaptiveMethod = 1, thresholdType = 1, blockSize = 11, and C = 2` is then applied. The resulting binary image is subjected to 15 iterations of dilation and erosion to reduce noise and ensure smoother contours. Contour detection is then used to identify the boundaries of the ROI. The bounding rectangle for each contour is cal-culated, and the largest of these is considered to contain the contour of the biopsy. The biopsy's contour is then fitted with the minimum area ("rotated") rectangle, as well as with a concave polygon to determine the ROI for the morphometry analysis.
The ROI detection is performed on the highest zoom level (lowest resolution) to achieve optimal performance. The rotated rectangle and concave polygons obtained at this stage are then scaled to the desired zoom level using bicubic scaling followed by simple thresholding (`threshold = 128`) to remove artifacts and produce binary images. The angle of the rotated rectangle is used to calculate the rotation required for the coarse pre-alignment.
The resulting images are exported at a resolution of 3000x7000, which corre-sponds to an 8.0x downsample from the lowest zoom level of the original slide, with a pixel ratio of 0.970136 µm. The downsampling factor can be decreased or eliminated, but this would result in a significant increase in computation time for downstream tasks”
R2: (3) What is the total amount of data required to train the U-Net model? As far as I understand, the data only involves the one patient mentioned in Chapter 2.1, so how can the diversity of data required to train the model be guaranteed? The paper only divides the training set and testing set, why does the author not consider the conventional train-val-test division, or the author uses cross-validation to train the model to avoid the ideal results brought by data leakage?
CA: The U-Net model is trained on 658 512*512 patches extracted from 7 fully annotated histological slides all from the same patient. From the overall methodology perspective, the choice of segmentation model is of little concern. In principle human annotators could also produce segmentation maps for each histological slide, albeit much slower. We have chosen U-Net as it was designed to work reliably with smaller datasets like ours. To validate that there’s no data leakage in the training process we have performed five-fold cross-validation as requested by the reviewer. The classification accuracy for the five folds are (in the chronological order of training the folds) 0.9548, 0.9565, 0.9304, 0.9608, 0.9607. We have revised the manuscript to elaborate on the cross-validation:
“These maps were split into tiles of size 512x512 pixels with a 64-pixel overlap in each direction and then divided into a five-fold cross-validation set up, meaning within each fold the model was trained on 80% of the data and validated on 20%. U-Net architecture [3945] was trained for the semantic segmentation task, modified from the original to include batch normalization in the 3x3 2D convolution blocks and dropout after the concatenations, to combat overfitting and improve the generalizabil-ity with limited training data, which is confirmed by the resulting validation accura-cies being 0.9548, 0.9565, 0.9304, 0.9608, 0.9607 for the five folds respectively.”
In case of a data leak between the folds would cause the accuracies to be monotonically increasing as the model overfits. The presented accuracies only show the variability in the validation sets. This is also supported by the attached loss curves. We do agree that if the study involved multiple patients the train-val-test division would be necessary to validate the inter-patient transferability of the segmentation model.
- 1. Figure: Loss curve for Fold 1
- 2. Figure: Loss curve for Fold 2
- 3. Figure: Loss curve for Fold 3.
- 4. Figure: Loss curve for Fold 4.
- 5. Figure: Loss curve for Fold 5.
R2: (4) The clarity of the figures and the texts in the figures needs to be improved, such as Figure 3 and Figure 4.
CA: We have updated the manuscript as requested, improving the clarity of the figures.
R2: (5) Table 1 is incomplete and may need to be reformatted.
CA: We have reformatted and completed Table 1.
R2: (6) Is it a preoperative or intraoperative bone assessment? As far as I understand, Chapter 1 wants to express the importance of preoperative evaluation, but in the experiment, it is an intraoperative evaluation (after the bone biopsy sample was taken, the implant placement procedure was performed). The author is invited to explain this part.
CA: The biopsy sample is harvested at the time of implant placement. However, because of the lengthy histological and computational procedures it could be considered a postoperative assessment technique. We have modified the manuscript to clarify this and added clinical implications as well.
We have added the following to the Introduction section:
“Postoperatively, microarchitectural evaluation of dental implant recipient bone by histological and microcomputed tomography (microCT) analysis of bone core biopsy samples allows the researcher to determine the risk factors of implant failure and to evaluate the clinical performance of bone graft materials. Histological examination of bone core biopsy samples harvested during dental implant placement may help identify underlying pathologies in case of implant failure.”
“A three-dimensional histological reconstruction that represents true trabecular bone microarchitecture would allow for a more comprehensive analysis of implant recipient bone. Such an assay may be utilized in the clinical research of bone graft materials and clinical pathology of implant failure as well.”
We have added the following to the Discussion section:
“However, correlation between two variables does not infer a valid method for measurement. Because of these difficulties in predicting bone quality, it is increasingly important to research recipient bone microarchitecture following bone augmentation procedures to determine whether factors such as healing time, bone graft materials used defect volume, characteristics of local anatomy, and systemic diseases or medication influence quality of augmented bone.”
“Such a three-dimensional histological reconstruction can be used in clinical and pre-clinical biomaterial research for postoperative evaluation of augmented bone. For the clinician, a comprehensive microarchitectural and histological assessment of implant recipient bone following dental implant placement may identify potential causes of implant failure.”
We would like to thank Reviewer 2 for their suggestions. Elaborating on the methodology, improving reproducibility, and presentation greatly elevates the quality of the manuscript. We hope that the reviewer finds the changes made sufficient to recommend our manuscript for publication.

Round 2
Reviewer 2 Report
Comments and Suggestions for Authors
Based on a thorough assessment of the author's revised manuscript and accompanying cover letter, I believe that the manuscript has effectively resolved and enhanced the areas that I previously raised. The author's revision efforts have significantly elevated the overall quality of the manuscript, and the appropriate modifications made to the research methods have brought forth a clearer research path. Moreover, the updated literature review has contributed to a more precise and comprehensive problem description, along with an introduction to the latest advancements in technology.
The author has made an effort to addressing my concerns, and as a result, the revised manuscript demonstrates noticeable improvements. I have no additional comments to provide.